# Effect of Winter Road Maintenance on the Asphalt Road Surface—Experience in Slovakia and the Czech Republic

**DOI:** 10.3390/ma15165618

**Published:** 2022-08-16

**Authors:** Silvia Capayova, Denisa Cihlarova, Petr Mondschein

**Affiliations:** 1Department of Transportation Engineering, Faculty of Civil Engineering, Slovak University of Technology, 810 05 Bratislava, Slovakia; 2Department of Transport Constructions, VSB-Technical University of Ostrava, 708 00 Ostrava, Czech Republic; 3Department of Road Structures, Czech Technical University in Prague, 166 36 Prague, Czech Republic

**Keywords:** asphalt pavement, bituminous binder, asphalt mixture, laboratory verification, physical-mechanical properties, winter road maintenance, de-icing salt, pavement assessment

## Abstract

The properties of asphalt mixtures and the quality of their execution are key factors for their service life and durability in the pavement structure. This research aims to study the effect of deicing chemicals (sodium chloride, NaCl) on asphalt mixtures in laboratory conditions to know the changes in properties when the samples were loaded with a different number of freeze–thaw cycles. The behavior of the standardly used asphalt mixtures and bitumen binders was verified by the exposure to sodium chloride solution (20 g/L NaCl). In the first part of the experiment, penetration, elastic recovery, and softening point tests were performed for paving grade bitumen 50/70 and 70/100, and polymer-modified bitumen PMB 25/55-60. Furthermore, asphalt concrete (AC 11) mixtures with different air voids were subjected to 25, 50, and 75 freeze–thaw cycles to determine the effect on the service life of the pavement structure. Following the Czech standard, CSN 73 6161, and the TP 170 regulation for the design of asphalt pavements, the Marshall test and the stiffness modulus were determined for all asphalt samples. The obtained results show a negative effect of freeze–thaw cycles on the properties of asphalt concrete surface course mixtures and bitumen.

## 1. Introduction

Asphalt is one of the most used construction materials on roads in Europe. Approximately 98% of pavements in Slovakia and the Czech Republic have asphalt mixtures in the covers. These roads are primarily in the rural area, with high traffic load, affected by climatic factors, and roads where winter maintenance is based on the application of chemical spreading materials. Several factors must therefore be considered when designing a road pavement. The first is the design of the type of pavement structure depending on the traffic load. Next is the choice of materials and technological procedures, from which the number of pavement layers and their dimensions is determined. When choosing materials for asphalt pavement, it is important to consider the temperature extremes and methods of road maintenance. Due to the viscous-elastic properties of bitumen, the asphalt mixture has a low viscosity at high temperatures and high viscosity at cold temperatures. The high temperature of pavement makes the asphalt softer; thus, the risk is high that heavy vehicles cause rutting due to the plastic deformation. As the ambient temperature decreases, the elasticity and stiffness of asphalt increase. At the same time, the relaxation capacity decreases. By preventing thermal shrinkage, so-called cryogenic tensile stresses are created during cooling, which at temperatures of −20 °C are so great that they exceed the tensile strength of the asphalt and lead to the formation of cracks near the surface. Furthermore, the cracks gradually penetrate downwards and reduce the service life of the pavement structure by the erosive action of the penetrating water. In winter, the covers undergo repeated stresses when the effects of freeze–thaw cycles and the effect of de-icing salt are added under constant traffic load.

It is known that the type of asphalt binder significantly affects the fatigue properties of the asphalt mixture. Some bituminous binders currently available on the Central European market contain a certain ratio of visbreaking residues, which, according to [1], affect the properties of the binders in middle and elevated service temperatures.

Worldwide, many research tasks are devoted to the issue of the impact of temperature extremes and related processes on individual activities and sectors. The reason is the risks related to climate change, which have been at the forefront of current global environmental risks for several years. Following the Global Risk Report 2021 [2], in the last ten years, these risks are beginning to appear in the top five, along with economic, social, and technological risks. In 2019, “extreme weather, climate action failures, and natural disasters” were in the top three. In 2020, only environmental risks were in the top five, in addition to “biodiversity loss and human-made environmental disasters”. Global warming forecasts point out that, by 2100, our planet could warm by an average of 3.5 to 7.5 °C. In Central Europe, and thus also in the Czech Republic and Slovakia, the average annual air temperature has risen by about 3.0 °C over the last 140 years. Significant changes are in the frequency of extremes—a rapid rise in extremes of maximum temperature and a rapid decrease in extremes of minimum temperature [3,4]. Manifestations such as strong winds, storms, torrential rains, and thunderstorms in the winter with a large volume of snow cover are more common. The consequences of these changes are immediate and intense and require adequate adaptation measures regulated in international [5] and national [6,7,8] documents.

These extreme weather events cause higher demands on the materials of the pavement structure layers and the construction technologies. A number of laboratory findings showed that the freeze–thaw cycle significantly causing a change in the load-bearing capacity of the pavement (load transfer) and the stiffness of materials (permanent deformation, fatigue) [9,10,11,12,13,14]. The demand for increasingly high-quality road construction material and the improvement of strength and deformation parameters of the mixture caused the use of various other materials in asphalt mixtures. An example is the use of various additives [14,15] that have been standardly used in several countries around the world since the 1990s or the application of materials such as crushed rubber, sulfur, which have been used in asphalt mixtures since the 1970s, which allows the long-term monitoring of their effect on the strength, fatigue life, and durability of different asphalt mixtures [13,16]. However, it is essential to consider the actual rationale for the use of the additive or technology. Even without the use of additional components, the asphalt mixture should be designed with sufficient parameters so that the layer and, consequently, the entire roadway can withstand the influencing factors throughout its lifetime.

There are several research tasks and programs in Central Europe that deal with the effect of increasing temperatures on the physical-mechanical properties of asphalt pavement materials. A few of them are aimed at assessing extreme negative temperatures or the increasingly frequent alternation of extremes of positive and negative temperatures. The studies [17,18] documented the behavior of asphalts at high and low temperatures, including the effect of freeze–thaw cycles. As confirmed by studies [19,20,21,22], the long-term application of sodium chloride (NaCl) has negative consequences on soils and the water environment. Heavy metals are released and chlorides increase. However, despite the above-mentioned negative effects of sodium chloride on the environment, it will continue to be used for its availability, applicability, and economic simplicity. Based on the mentioned experience, it is assumed that the repeated effects of repeated freeze–thaw cycles and saline solution on the strength and deformation characteristics are negative. However, according to the authors [23] and their experience, salt water shortens the time the mixture is in contact with frozen water, which is one of the most harmful effects on asphalt. When the mixture is exposed to cycles, the saline solution has a protective effect on the samples that remain immersed in it. Salt in water protects the bituminous putty, maintains mechanical strength, and increases the number of load cycles for any range of stresses. The results of their research point to the fact that temperature has a detrimental effect on the mechanical properties, but samples immersed in salt water achieve better results than their analogs, which are immersed in distilled water. However, it must be emphasized that the authors used only five freeze–thaw cycles, which may be a number characteristic of Spain, but not suitable for conditions in the Czech Republic and Slovakia.

In general, current issues of road de-icing adjustments in former Czechoslovakia have been more actively addressed at various research institutes since the 1970s. The questions are mainly concerned with the necessary development of chemical spreading materials with increased efficiency and more favorable environmental impacts. In addition to standard materials, research was also carried out on sources of cheap secondary raw materials. The most extensive research program on improving winter maintenance with chemicals was carried out in 1978 at the Transport Research Institute in Žilina (Žilina, Slovakia) and the branch in Brno (Brno, Czech Republic). The issues of winter maintenance overlapped with the maintenance programs of other professional and scientific workplaces in Žilina and Bratislava, as well as with the activities of motorway maintenance centers and district road administrations. From the 1970s to the 1990s, reports on the effect of chemical sprinklers on road surfaces, road equipment, bridge structures, reports specifying criteria for the protection of roadways against the effects of freeze–thaw cycles, etc. have been prepared [24]. An extensive information base and working materials from world road congresses were developed, which made it possible to shape the direction of progress in the modernization of applications of chemical de-icing agents. Whereas in the past the production of de-icing agents was only possible through the importation of chloride compounds, in the last 25 years, there have been several possibilities for the production of chloride de-icing agents even at home. Very interesting for the decision-making process can be the computer decision tree for the application of de-icing agents [25], which could increase the efficiency of winter maintenance processes in our conditions.

The authors aim to point out the need to introduce and implement freeze–thaw cycles with the action of salt solution during standard asphalt testing in the conditions of both countries. The issue of the impact of winter maintenance on the physical and mechanical properties of asphalt has not received much attention in Slovakia for a long time. It is not included in the standards and technical regulations for testing asphalt. However, in the period before the adoption of EU standards, freezing cycles were also used in some tests and, in general, more attention was paid to the impact of chemical de-icing materials on asphalt pavements [26]. Currently, the resistance to freeze–thaw cycles is obligatorily assessed only for concretes and cement-bonded mixtures. The situation is slightly different in the Czech Republic, where the CSN 73 6161 standard for determining the adhesion of asphalt binders to aggregates defines this property even under the conditions of freeze–thaw cycles in NaCl solutions.

The next part of the article is devoted to the analysis of the legislation of both states in the given issue, which is also one of the important bases for setting up the experimental part. The setup of the entire experiment was based on the mentioned Czech standard CSN 73 6161 and TP 170—regulation for the asphalt pavement design, as well as, due to the very similar topographic and climatic conditions of both countries and the mutual long-term existence of common standards and technical regulations during the existence of a common state. The results of the laboratory measurements were the basis for drawing conclusions and further recommendations.

## 2. Materials and Methods

### 2.1. Legislation and De-Icing Operations

De-icing is a traditional snow and ice control application. It requires the maintenance personnel to wait until snow accumulates on the road before starting to plow the pavement surfaces and clean the highway with de-icers and/or abrasives.

Different from de-icing operations, anti-icing is defined by the ice and snow control application of preventing the formation and/or development of strongly bonded ice and snow by the timely practices of freezing-point-depressant chemicals. This is a method that takes preventive snow/ice control strategies. Brine decay was studied on highways ATH-50 PCC and PICK-23 AC versus time and traffic order to design effective winter maintenance [25].

In the Czech and Slovak Republics, deficiencies in road possibility are eliminated by the responsible road administrators, by sprinkling with inert, chemical, or mixed materials or by the mechanical removal of snow and ice.

In Slovakia, winter maintenance and its scope are in valid technical regulations. Following Act no. 135/1961 On roads (Road act) as amended, two technical conditions applied until June 2022: TP 039, Use of magnesium chloride-based spreading materials on roads, and TP 040, Use of sodium chloride-based spreading materials on roads. Currently, one regulation has been in force since July 2022: TP, 039 Use of spreading materials for winter maintenance on roads [27]. The following are used as de-icing agents: sodium chloride, calcium chloride, a mixture of sodium and calcium chloride, soladin, tonacal, and urea (carbamide). Sodium chloride-based NaCl-spreading materials or solutions with a temperature range of action up to −15 °C are mainly used under normal conditions. Under exceptional conditions, they can also be combined with other chemical spreading materials, such as CaCl_2_ or MgCl_2_. Especially under exceptional conditions and at mountain passes, it is recommended to use materials based on magnesium chloride MgCl_2_ with a temperature range of action down to −34 °C [28]. The specific requirements are summarized in Table 1 (for regulations valid until June 2022 and from July 2022).

The issue of winter maintenance in the Czech Republic is regulated in Decree of the Ministry of Transport and Communications No. 104/1997 Coll., implementing the law on road traffic, as amended. According to the order of importance, winter maintenance alleviates defects in the possibility of roads caused by weather conditions in the winter. The decree determines when and in what order the roads are maintained and the dosing of chemical spreading material due to weather conditions. In the Czech Republic, sodium chloride is used to sprinkle roads, motorways, and local roads, except non-motorized roads, which is effective for removing ice and snow at temperatures up to −5 °C (under certain conditions even lower), or calcium chloride CaCl_2_ with temperature efficiency below −15 °C. Material dosing requirements are in Table 2. Differences in spreading salt parameters for Slovakia and the Czech Republic are in Table 3.

The regulations of both states are similar, and the differences are listed in Table 1, Table 2 and Table 3. The total consumption of spreading salts must not exceed 80 g∙m^−2^ on one intervention day in Slovakia and 60 g∙m^−2^ in Czech Republic. Following TP 039 [27], materials based on sodium chloride have a certain proportion of K_4_Fe(CN)_6_ anti-caking additive, which is not specified in Czech regulations. The Slovak regulations also contain limits for the content of heavy metals and unsuitable admixtures. These have not yet been set in regulations in the Czech Republic.

The grain size distribution is essential for the method of use and the spreading technique. Fine particles cause rapid surface dissolution, but the effect in depth is limited. Coarse particles penetrate more into the depth. The snow covers or thicker icing is crushed by the effect of road traffic. Upon the delivery of salt, information on the chemical composition, the proportion of insoluble admixtures, the moisture content, and the grain size distribution must be documented.

Chemical de-icing agents are used in crystalline form or the form of solutions (brine) with the desired concentration. Mixing stations are intended for their preparation. Direct use of de-icing solutions to remove icing or snow is possible if temperatures below −3 °C are not expected to fall. They are applied with sprinklers, which allow even spraying. The dose limits for the solutions are based on the concentration of the active substance, the amount of which must not exceed the above dose limits.

Czech standard No. CSN 73 6161; Annex B Determination of the adhesion between asphalt binders and aggregates specifies the method for testing the frost resistance on freezing and thawing compacted asphalt mixtures in the NaCl environment [29]. The effect of NaCl agents on asphalt mixtures is not defined in the standards of the Slovak Republic. Furthermore, the standard EN 12697–41 is in force in the territory of both states, which specifies a test method for determining the resistance of asphalt mixtures to de-icing fluids (acetate and formate solutions).

### 2.2. Experimental Methods

The aim was to find out the effect of repeated freeze–thaw cycles with the action of water and sodium chloride solution on the strength and deformation characteristics of standard used asphalt concrete mixtures. The laboratory tests were chosen based on the requirements of CSN 73 6161 (Marshall test) and TP 170 [30]—the asphalt pavement design method of the Czech Republic (Stiffness modulus). Even in the Czech and Slovak regulations, standards are not defined for the requirements for testing the asphalt binder under the action of freeze–thaw cycles and salt solution. The basic and, at the same time, the simplest tests for determining the change in binder properties, penetration, softening point, and elastic recovery tests were chosen. The aim was to find out whether the salt solution affects a given property at all, and whether the binder will harden and become brittle. The aim was not to determine the low-temperature and rheological properties. The binder testing procedure was based on the experience from Germany and Scandinavia and the experience with experts in the field [31].

For the experiment itself, it was necessary to simulate conditions as realistic as possible. This also applied to the design of the number of freeze–thaw cycles. The average number of frost days per year (days when the temperature drops below 0 °C during 24 h) in Slovakia is 115 days [28,32], as a long-term average for the 1931–1990 period. In the Czech Republic, it is 112 days [33], as a long-term average for the 1961–1990 period. Due to the diversity of the two countries, the number of frost days within the republics is different. The lowlands have on average 90–100 frost days, but it can be 160 frost days in the mountain valleys. After 1990, due to general warming, all characteristics shifted by about 1 °C.

On average, 13 freeze–thaw cycles occur in the Czech Republic per year. As Slovakia has roughly the same number of frost days, the probability of the number of frost–thaw cycles is the same. According to the Czech standard, the number of 25 cycles is specified for testing the frost resistance on freezing and thawing compacted asphalt mixtures. According to the above, it corresponds to two years of road use. Usually, the surface course is replaced after about 8 to 12 years. For this reason, the test was extended by 50 and 75 freeze–thaw cycles.

#### 2.2.1. Bituminous Binders

Samples of standard used paving grade bitumen 50/70 and 70/100, and polymer-modified bitumen PMB 25/55-60 for testing penetration [34], elastic recovery [35], and softening point [36] were placed in sodium chloride solution and placed in an oven at 40 °C for 2 and 7 days. This test temperature was chosen to accelerate the action of the salt solution on the binder. The reference samples were placed in an oven without a salt solution. Thus, the results were not affected due to the different aging of the binder.

#### 2.2.2. Asphalt Mixtures

For testing the frost resistance on freezing and thawing compacted asphalt mixtures, Marshall samples were made from ACO 11 + asphalt mixture (asphalt for surface course, D_max_ = 11 mm). The Standard prescribed 25 freeze–thaw cycles, 50 cycles and 75 cycles were added. Subsequently, the samples were divided according to the average bulk density into four groups: air, 25 freeze–thaw cycles, 50 freeze–thaw cycles, and 75 freeze–thaw cycles. The absorption of all samples was determined; see Figure 1. Then, the samples were subjected to a given number of cycles of freezing (minimum 4 h at −20 °C ± 2 °C) and thawing (minimum 4 h at room temperature +20 °C ± 2 °C, in saline solution); see Figure 2.

#### 2.2.3. Saline Solution

According to CSN 73 6161—Annex B [29], a saline solution in a concentration of 20 g/L was produced. The solution was made from distilled water and de-icing salt. It is a solid, odorless, white crystalline substance—sodium chloride NaCl, intended for winter maintenance but also direct consumption in the food industry. It is not a dangerous chemical. It is a commonly used as standardized sprinkling salt in the territory of the Slovak Republic and the Czech Republic. Its basic properties are listed in Table 4.

## 3. Results

### 3.1. Asphalt Binder Properties

The test results of asphalt binders [34,35,36] are in the graphs of Figure 3, Figure 4 and Figure 5. The values of the penetration and softening point of the 50/70 and 70/100 paving grade bitumen and PMB 25/55-60 polymer-modified bitumen follow the requirements of the standards. For the 50/70 binder, the softening point corresponded to the requirement of the standard, but the detected penetration was lower by two penetration units. However, this inconsistency does not affect the experiment.

The results’ analysis of the empirical characteristics of asphalt binders showed that the change in penetration occurs after the first two days in the saline solution. For the 70/100 binder, the penetration of the new binder and the binder two days in saline solution decreased by six penetration units, i.e., the change was about 8%. For the 50/70 binder, there was a decrease of four penetration units, i.e., also the change was about 8%. For a polymer-modified bitumen, this decrease is minimal. It was a change of one penetration unit, i.e., the change was about 3%. For all three binders, the further effect of the saline solution did not change the penetration. The penetration after seven days in the solution was identical to the penetration after two days in the solution.

The softening point results have different behavioral tendencies. Gradually, there is a slight increase in the softening point between the second and seventh day of the effect of the salt solution. For the binder 70/100, the total change was 1.1 °C and 1.2 °C, respectively, a change of 2.5%. For the binder 50/70, there was an increase of 2.3 °C, a change of 4.4%. For the PMB binder, in one case, there was an increase of 0.6 °C and a decrease of 0.8 °C in the other case. It can be stated that the saline solution does not affect this property of the PMB binder.

Elastic recovery was determined only for the PMB binder. After two days in the saline solution, there was a decrease of 3.1%, after another five days a further 2.5%. The total change in elastic recovery was 5.6%, i.e., the change was about seven percentage points.

In conclusion, the evaluation of the characteristics of asphalt binders revealed that there is a maximum change of 8% in the properties after seven days in the saline solution.

Additionally, in Figure 6, the change in the shape of the test samples (elastic recovery test) due to the effect of saline solution can be seen.

### 3.2. Strength and Deformation Characteristics of Asphalt Mixtures

Marshall samples were into four groups divided: air (reference: 0 freeze–thaw cycles), 25 freeze–thaw cycles, 50 freeze–thaw cycles, and 75 freeze–thaw cycles. All four groups had approximately the same bulk specific gravity (Figure 7). Air void value was 2.0% for the “Air” and “75 freeze–thaw cycles” groups of samples and the value of 2.2% for the “25 and 50 freeze–thaw cycles” groups of samples. The increasing number of freeze–thaw cycles increases the absorption value from 1.5% to 3.4% (Figure 8): “Air” group samples—values of 1.1–1.9%; “25 freeze–thaw cycles”—values of 2.3–2.4%, “50 freeze–thaw cycles”—values of 2.8–3.3%; and “75 freeze–thaw cycles”—values of 2.9–4.2%. Relationship between the absorption of asphalt and the number of freeze–thaw cycles is in Figure 9. Marshall stability, flow value, and Marshall stiffness [38] (Figure 7 and Figure 10) were measured, and the change in stiffness modulus [39] is shown in Figure 11, Figure 12, Figure 13 and Figure 14. The results of both tests show identical trends. The increasing number of freeze–thaw cycles negatively affects the deformation characteristics of the asphalt mixtures. The Marshall stability decreased from the original value of 10.8 kN to 9.07 kN due to freeze–thaw cycles, which is a decrease of about 16%. The flow value has the opposite trend. The Marshall stiffness decreased from the original value of 3.13 kN/mm to 2.00 kN/mm. It is a decrease of about 36%.

Figure 12 and Figure 13 show the relationship between the decrease in stiffness modulus [38] and the number of freeze–thaw cycles at the test temperatures of 15 °C and 40 °C. The trend of results is the same. As the number of cycles increases, the stiffness modulus decreases. From the basic value of 10,219 MPa, after 75 freeze–thaw cycles, there was a decrease of about 32% of the stiffness modulus at 15 °C and a decrease of about 31% at a temperature of 40 °C. The relationship between the number of freeze–thaw cycles, the absorption, and the stiffness modulus show Figure 14. When exposed to chemical de-icing agents, it can be assumed that a higher number of freeze–thaw cycles applied will result in higher absorption and thus lower strength and deformation parameters. As can be seen, for samples loaded with 0 freezing cycles, there are absorption values with a maximum value of approx. 1.5% and the average value of the stiffness modulus is 10,219 MPa (15 °C) and 979 MPa (40 °C). After the application of 75 freeze–thaw cycles, there is an increase in absorption to an average value of 3.4% and a decrease in the stiffness modulus of 68% for both temperatures. Such a relationship can also be assumed, in general, for other types of asphalt mixtures.

### 3.3. Application of Results—Pavement Assessment

The obtained data can be used to assess the pavement structure and evaluate the impact of winter chemical maintenance on shortening the theoretical life of the pavement structure. Two pavement structures were assessed using the TP 170 methodology [30]. The influences of the decrease in stiffness modulus over time due to the effect of salt on the pavement surface course (theoretical service life) was determined. It is a certain simplification. Firstly, in the sense of the effect of chemical de-icing agents on the asphalt pavement layer only from the upper surface. Second, the binder ages. This can compensate for the decrease in stiffness modulus due to chemical de-icing agents.

Two catalogue pavement structures were selected for illustration: a motorway pavement structure (DO-N-1-PIII) and a second-class road pavement structure (D1-N-2-PIII) (Table 5). The used asphalt mixtures are according to different quality requirements. For No. 1, asphalt concrete AC “S” was used, with mixtures with increased resistance to permanent deformations; for the highest traffic load classes (TLC) with an average daily intensity of more than 7500 heavy goods vehicles (HGVs); for wearing and binder course compacted in the laboratory design by 2 × 75 blows in Marshall compactor and 2 × 50 blows for the use in the base course. For No. 2, asphalt concrete “+” was used, with mixtures compacted in the laboratory design by 2 × 50 blows of Marshall compactor; for wearing and binder course for TLC II to IV (HGV = 101–3500) and for a base course with TLC V to VI (HGV = 15–100).

Only an 8-year period was assessed using the TP 170 methodology, for which data on the decrease in stiffness modulus depending on the number of freeze–thaw cycles were available. The reference structures were those for which a constant value of the surface course stiffness modulus was assumed for the whole eight years of the assessment period. This is a value that was experimentally determined on test samples not affected by saline solution.

The TP 170 methodology evaluates the theoretical life of a road structure by relative failure of D_cd_. Every load generates a stress (relative strain or stress) in the structure. In the compacted layer, depending on the load intensity, the failure of the layer occurs, while in the subsoil, permanent deformation occurs. The accumulation of failures and permanent deformations leads to pavement failures. The reliability of the design of the pavement structure must be appropriate to the traffic volume and the traffic significance of the road, which is based on Miner’s hypothesis, i.e., the accumulation of partial damages is linear, the damage is cumulative, and the running order does not matter. Pavement damage occurs when the sum of the partial relative failure is D_cd_ ≤ 1. The total relative failure is then given by the superposition of the partial relative strain:(1)$$D_{\mathrm{cd}\;}=\sum_{i=1}^{m_{i}}\sum_{j=1}^{m_{j}}D_{\mathrm{ij}}$$
where D_cd_ presents the total relative failure during the design period, m_i_ presents the number of different categories of load sets, and m_j_ presents the number of different conditions.

In principle, this is the ability of the road structure to transfer traffic loads. This is expressed by the maximum horizontal relative deformation at the asphalt critical layer and by the vertical relative deformation at the subsoil of the road structure. Both deformations are compared with the effects of heavy vehicles over the entire theoretical life of the road. Table 6 and Table 7 show the change in stiffness modulus due to freeze–thaw cycles and due to chemical de-icing agents. A comparison of the assessment of the reference pavement structure with pavement structures No. 1 and 2 leads to the following conclusions:The length of the assessed period was eight years, and it corresponds to the average durability of the wearing course, which is stressed by 75 freeze–thaw cycles;The use of chemical de-icing salt causes a decrease in the stiffness modulus of the wearing course, and the service life of road structures is shortened;The service life of the road structure 1 is shorter by 5% for the asphalt-bonded critical layer during the eight-year design period, and by 11% for the subsoil; with a traffic intensity of 10,000 design trucks in 24 h, this is a reduction in the number of design trucks crossings for the assessed period of 8 years in the number of 1,460,000 HGVs for asphalt layers and 3,212,000 for subsoil;The service life of the road structure 2 is shorter by 7% for the asphalt-bonded critical layer during the eight-year design period, and by 12% for the subsoil; with a traffic intensity of 440 design trucks in 24 h, this is a reduction in the number of design trucks crossings for the assessed period of 8 years in the number of 89,936 design trucks for asphalt layers and 154,176 for subsoil.

## 4. Conclusions

The article [40], published in 1967, states that salt does not affect the change in the strength characteristics of asphalt mixtures. The results of this our study did not confirm this statement. The mentioned article does not state how many freeze–thaw cycles were performed before the Marshall test and what kind of solution was used at 25 freeze–thaw cycles, so the conclusions of both studies are not identical. The salt solution has the effect of changing the properties of asphalt binders and asphalt mixtures. The results of the characteristics of asphalt binders after seven days of stress in saline solution show a maximum change of 8%in the properties. The change in penetration occurs mainly after the first two days of stress, and the next stress is a minimal change in penetration. Between the second and seventh days of stress, a small increase in the softening point was recorded for 50/70 and 70/100 binders, and there was almost no change for PMB 25/55-60 binder. The final change in the elastic recovery of PMB 25/55-60 binder was 5.6%. The investigation of the behavior of asphalt binders was only marginal. The focus was on basic tests, which provided only basic information about the effect of salt on the binder. One of the next steps in the future will be to pay attention to low-temperature properties and rheological tests, which will provide a truer picture of the actual behavior of how the salt solution and repeated freeze–thaw cycles can affect the binder.

The decrease in strength and deformation characteristics occurs significantly after 50 and 75 freeze–thaw cycles. The increasing number of freeze–thaw cycles negatively affected the deformation characteristics of the asphalt mixture. Due to the freeze–thaw cycles, the Marshall stability decreased by about 16%, from the original value of 10.8 kN to 9.07 kN, and the Marshall flow value increased by about 30%—from the original value of 3.47 mm to 4.50 mm. The stiffness modulus decreased by about 32% at 15 °C, from the original value of 10,219 MPa to 6800 MPa, and the stiffness modulus decreased by about 31% at 40 °C, from the original value of 979 MPa to 666 MPa.

The conclusions from the stiffness modulus test show a relatively strong dependence between the absorbency of the test specimen, its voids, and stiffness. The high void content of the wear layer can negatively affect the service life of the road structure or the durability of the wear layer in localities where there is a more frequent fall of snowfall and repeated chemical winter maintenance. By expressing the relative failure of D_cd_, it has been shown that, with decreasing strength characteristics, the service life decreases. These declines are significant and must not be forgotten.

The results of this study may be the basis for a change (decrease) in the air void requirement for the wear layers. It would be very interesting to perform a similar study on open asphalt mixtures, which can reduce rolling noise. Their high void content increases the consumption of salt, which has a higher negative effect on the deformation characteristics of asphalt mixtures and reduces their durability.

## Figures and Tables

**Figure 1 materials-15-05618-f001:**
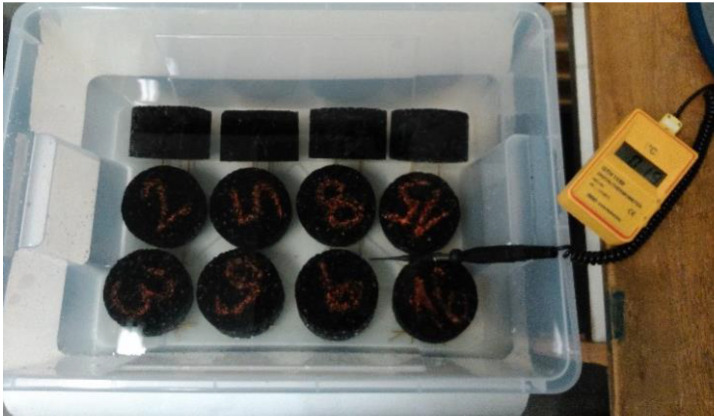
Samples in distilled water before freeze–thaw cycles (Photo: Authors).

**Figure 2 materials-15-05618-f002:**
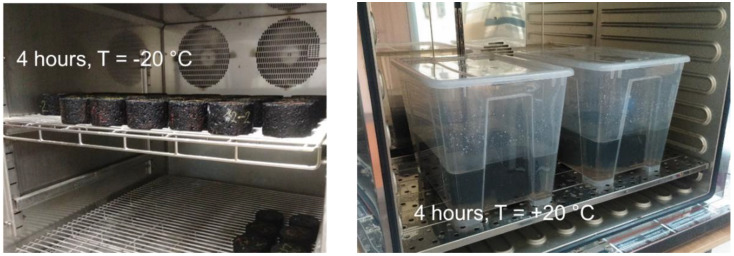
Freeze–thaw cycles (Photo: Authors).

**Figure 3 materials-15-05618-f003:**
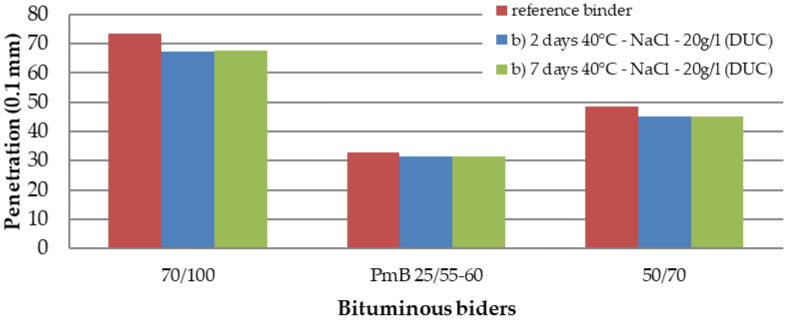
Penetration value (0.1 mm) depending on the duration of action of sodium chloride.

**Figure 4 materials-15-05618-f004:**
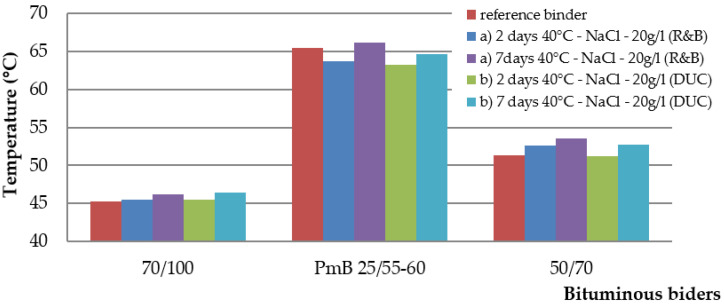
Softening point (°C) depending on the duration of action of sodium chloride.

**Figure 5 materials-15-05618-f005:**
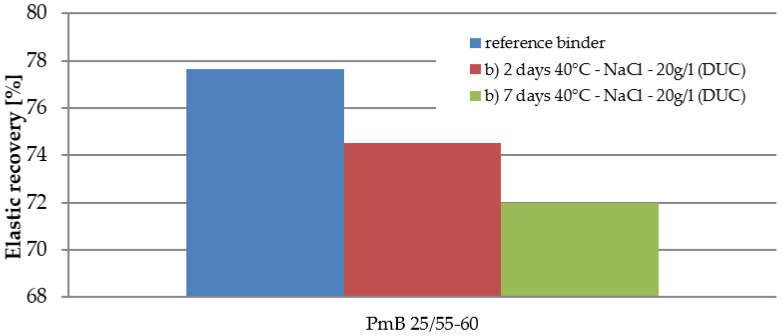
Elastic recovery (%) of the polymer-modified binder by the action of sodium chloride.

**Figure 6 materials-15-05618-f006:**
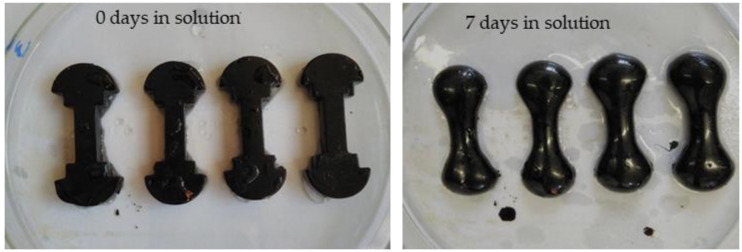
Change in the shape of the asphalt binder test samples (Photo: Authors).

**Figure 7 materials-15-05618-f007:**
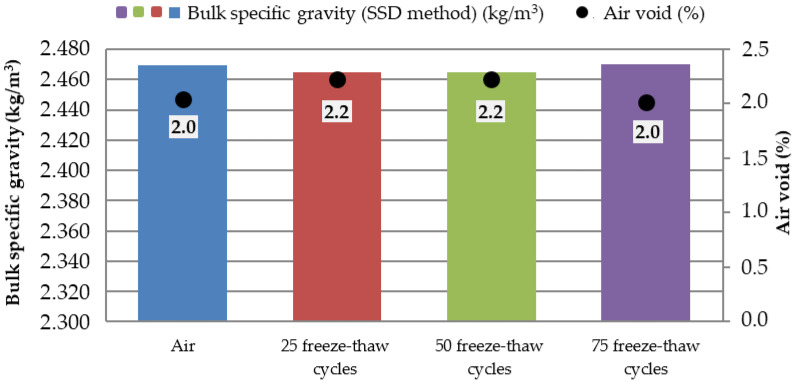
Bulk specific density and air void of the ACO 11 + mixture depending on the number of freeze–thaw cycles.

**Figure 8 materials-15-05618-f008:**
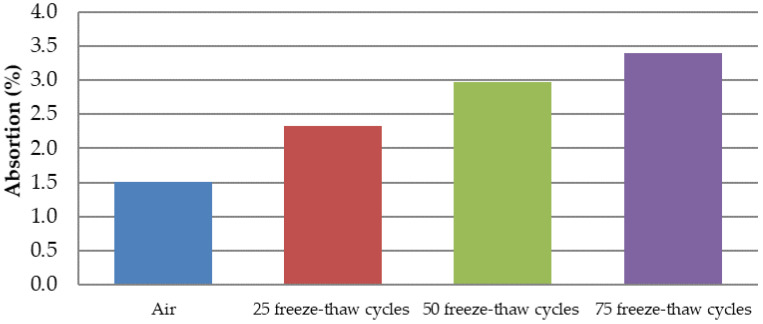
Absorption (%) of the ACO 11 + mixture depending on the number of freeze–thaw cycles.

**Figure 9 materials-15-05618-f009:**
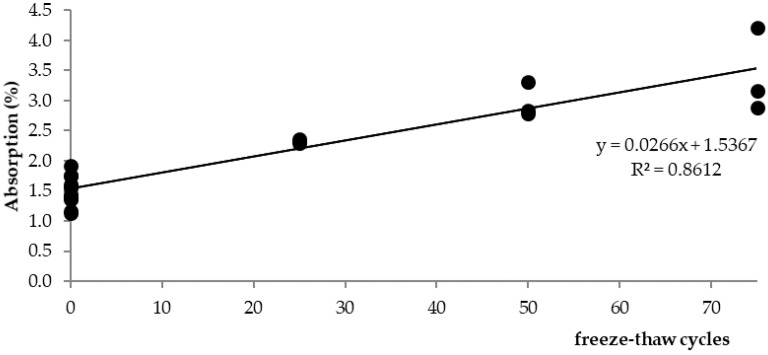
Relationship between the absorption of the mixture and the number of freeze–thaw cycles.

**Figure 10 materials-15-05618-f010:**
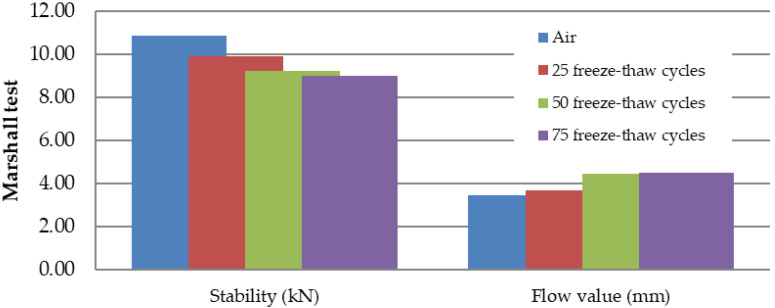
Marshall stability (kN) and flow value (mm) depending on the number of freeze–thaw cycles.

**Figure 11 materials-15-05618-f011:**
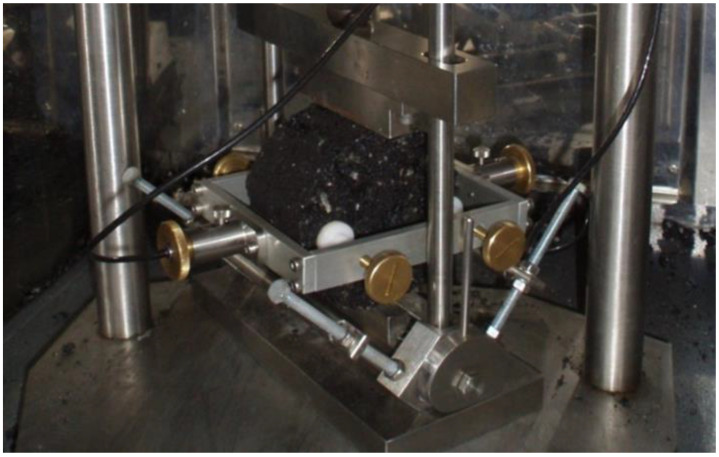
Stiffness modulus test (Photo: Authors).

**Figure 12 materials-15-05618-f012:**
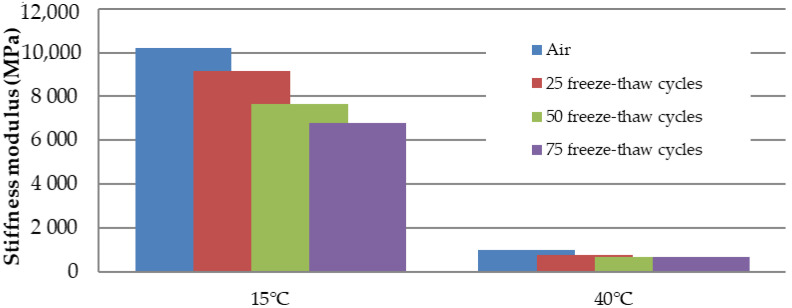
Stiffness of asphalt (MPa) depending on the number of freeze–thaw cycles.

**Figure 13 materials-15-05618-f013:**
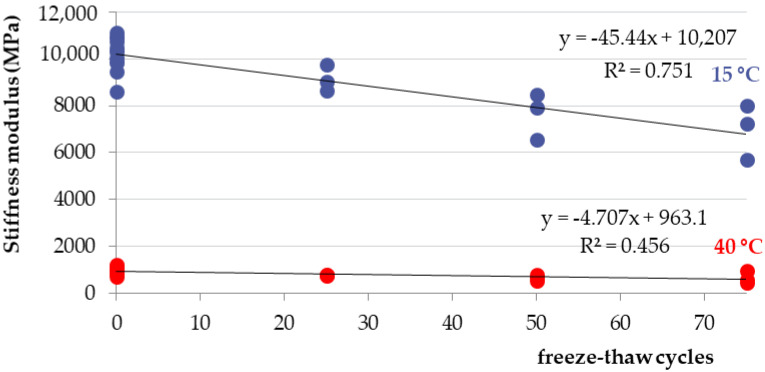
Relationship between stiffness modulus at the test temperatures of 15 °C and 40 °C and the number of freeze–thaw cycles.

**Figure 14 materials-15-05618-f014:**
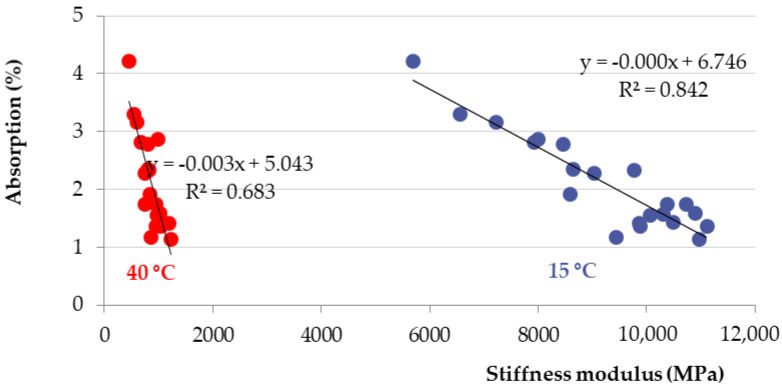
Relationship between stiffness modulus at the test temperatures of 15 °C and 40 °C and absorption of test samples.

**Table 1 materials-15-05618-t001:** Recommended dosage of chemical de-icing materials in Slovakia.

Conditions	Until June 2022			From July 2022		
	Dosage (g/m^2^)	NaCl	CaCl_2_	Dosage (g/m^2^)	NaCl	CaCl_2_
Icing up to 2 mm	20	to −5 °C	to −15 °C	20	to −7 °C	to −15 °C
Snow layer up to 3 mm	20			20–40		
Icing over 2 mm	40			40		
Extraordinary conditions	40–60			40–60		

**Table 2 materials-15-05618-t002:** Recommended dosage of chemical de-icing materials in the Czech Republic.

Conditions	Dosage (g/m^2^)	NaCl	CaCl_2_
Icing up to 2 mm	20	to −5 °C	to −15 °C
Snow layer up to 3 mm	20		
cing over 2 mm	40		
Extraordinary conditions	40–60		

**Table 3 materials-15-05618-t003:** Parameters of spreading salt.

Parameter	Slovakia	Czech Republic
The concentration of solution in chemical spreading (%)	20–25	18–21
Content of active solvent (%)	98	96
Content of dust particles (%)	max. 15	max. 5
Water content (%)	max. 2	max. 2
The proportion of anti-caking agent (g/kg)	10 to 200	-

**Table 4 materials-15-05618-t004:** De-icing salt properties [37].

Physical and Chemical Properties	Value
NaCl	≥97%
pH value at 20 °C (solution 100 g/1 L H_2_O)	7.5–8.5
Density	21,615 kg/dm^3^
Solubility (at 20 °C) in fats	360 g/L
Solubility (at 20 °C) in water	not soluble
Melting point (°C)	801 °C
Boiling point (°C)	1460 °C
Flash point (°C)	-

**Table 5 materials-15-05618-t005:** Assessed pavement structures.

		Pavement Structure No. 1		Pavement Structure No. 2	
		DO-N-1-PIII		D1-N-2-PIII	
		Traffic load class	S	Traffic load class	III
**Pavement structure layer**	wearing course	ACO 11 S	40 mm	ACO 11 +	40 mm
	binder course	ACL 22 S	80 mm	ACL 16 +	60 mm
	base course	ACP 22 S	150 mm	ACP 16 +	50 mm
		Unbound gravel	200 mm	Unbound gravel	150 mm
	subbase course	Unbound gravel	250 mm	Unbound gravel	150 mm
		Total	620 mm	Total	450 mm

**Table 6 materials-15-05618-t006:** Assessment of pavement structure No. 1 and relative failure of D_cd_ (change in the modulus of stiffness of wearing course due to the number of freeze–thaw cycles).

Number of Freeze–thaw Cycles		0	25	50	75
Design Period (Years)		2	4	6	8
Reference pavement structure	Relative failure of the asphalt-bonded critical layer	0.015	0.030	0.045	0.060
	Relative failure of the subsoil	0.016	0.032	0.048	0.064
Pavement structure No. 1	Relative failure of the asphalt-bonded critical layer	0.015	0.030	0.046	0.063
	Relative failure of the subsoil	0.016	0.033	0.052	0.072

**Table 7 materials-15-05618-t007:** Assessment of pavement structure No. 2 and relative failure of D_cd_ (change in the modulus of stiffness of wearing course due to the number of freeze–thaw cycles).

Number of Freeze–thaw Cycles		0	25	50	75
Design Period (Years)		2	4	6	8
Reference pavement structure	Relative failure of the asphalt-bonded critical layer	0.026	0.052	0.078	0.104
	Relative failure of the subsoil	0.034	0.064	0.096	0.136
Pavement structure No. 2	Relative failure of the asphalt-bonded critical layer	0.026	0.053	0.082	0.112
	Relative failure of the subsoil	0.034	0.070	0.111	0.155

## Data Availability

Not applicable.

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
