# Peer review of "Effect of Winter Road Maintenance on the Asphalt Road Surface—Experience in Slovakia and the Czech Republic"

_materials, 2022, doi:10.3390/ma15165618_

Round 1
Reviewer 1 Report
Manuscript ID: materials-1782980
Manuscript Title: Effect of Winter Road Maintenance on the Asphalt Road Surface - Experience in Slovakia and The Czech Republic
General comment:
This work introduced the influence of winter maintenance (using chemical deicing salt) on asphalt pavements. The laboratory studies were compressively done for obtaining the conclusions. It can be accepted after all my comments are well addressed.
Specific comments:
1. Abstract. The discussion on the test results could be further simplified, the introduction of the main findings should be enough.
2. The objective of the work is not well presented and clear to determine the novelty of the work.
3. The fitted formula needs to be checked again, such as Figure 16.
4. The quality of figures is very poor. Please change with high quality and resolution figures.
5. There are 16 figures in this manuscript (too much for one manuscript). Please select the most important one to be put in the manuscript. Alternatively, the authors can make like this; For example, Figure 9 (a, b and c). Please revise those figures accordingly.
6. The conclusions could be further reorganized.
Author Response
In accordance with all reviews, the article has been modified and edited. The abstract, research part and conclusions were edited in such a way that the objective, method of the experiment, results, conclusions, and recommendations were clearly defined. The order of references has also been adjusted.
Authors' answers and comments:
- Abstract. The discussion on the test results could be further simplified, the introduction of the main findings should be enough. Answer: Modified
- The objective of the work is not well presented and clear to determine the novelty of the work. Answer: Modified/Added
- The fitted formula needs to be checked again, such as Figure 16. Answer: Formulas revised and figures updated
- The quality of figures is very poor. Please change with high quality and resolution figures. Answer: All graphic attachments in Word and PDF look to us to be of sufficient quality. It is not a problem to provide graphs in Excel and photos in better resolution for the final version of the article.
- There are 16 figures in this manuscript (too much for one manuscript). Please select the most important one to be put in the manuscript. Alternatively, the authors can make like this; For example, Figure 9 (a, b and c). Please revise those figures accordingly. Answer: The numbering of the figures was corrected and their number adjusted.
- The conclusions could be further reorganized. Answer: Modified
Reviewer 2 Report
The binder tests used, penetration and softening point, are not a true representative of the rheological or even physical properties of the binder as they are only performed at room temperature (25 degrees Celsius)
Low-temperature evaluation of the binder is required for this particular study.
Stiffness modulus is an obsolete test, along with Marshall's stability and flow test. These can not be used for pavement evaluation
Author Response
In accordance with all reviews, the article has been modified and edited. The abstract, research part and conclusions were edited in such a way that the objective, method of the experiment, results, conclusions, and recommendations were clearly defined. The order of references has also been adjusted.
Authors' answers and comments:
1. The binder tests used, penetration and softening point, are not a true representative of the rheological or even physical properties of the binder as they are only performed at room temperature (25 degrees Celsius). Low-temperature evaluation of the binder is required for this particular study.
Answer: Even in the Czech and Slovak regulations and standards are not defined the requirements for testing the asphalt binder under the action of freeze-thaw cycles and salt solution. The binder testing procedure was based on experience from Germany and Scandinavia and experience with experts in the field. As the basic and, at the same time, the simplest tests for determining the change in binder properties, penetration, softening point, and elastic recovery tests were chosen. The aim was to find out whether salt solution affects the given property at all, and whether the binder will harden and become brittle. The aim was not to determine the low-temperature and rheological properties.
2. Stiffness modulus is an obsolete test, along with Marshall's stability and flow test. These can not be used for pavement evaluation.
Answer: Valid standard ÄŒSN 73 6161, Annex B prescribes the determination of the resistance of the asphalt mixture against freeze-thaw cycles in salt solution using the Marshall test with 25 cycles. The valid regulation TP 170 prescribes formulas for the assessment of asphalt pavements, in which it is necessary to insert the stiffness modulus values.
Reviewer 3 Report
The work presented in the manuscript is very interesting and relevant in road maintenance process. Issues facing the de-icing the roads is once of the greatest challenge for road asset owners. Authors have carried out very interesting work however the manuscript reads as a report rather dan scientific paper. Authors should rewrite (restructure)- the manuscript following the journal template.
My additional comments are as follows:
1. The English throughout the paper requires moderate corrections.
Eg. Abstract, pg 1, ln 20: we made many tests.. – awkward writing I am suggesting: # of tests were carried out ….
2. Introduction paragraphs 1 and 2, here authors introduce risks and global climate challenges without a context, authors should present the challenge that they are addressing at start of the Introduction then this would make more sense as it is it reads discussion on the climate change rather than issue that authors are addressing.
3. Pg. 3, ln 91 – Authors state: “Asphalt binder is known to play an important role when determining fatigue properties …..” Asphalt can’t determine asphalt fatigue properties. A ITFT or 4PBT us used to determine the fatigue properties, asphalt is material used to improve fatigue properties of the pavement…
4. Pg 3 ln 100 – 103: Authors Refer to research programme carried out in Slovakia, please reference publicly available works that describe this work.
5. Pg 3, ln 112: computer decision tree [12] – please include schematic diagram of this tree and explain how it works
6. On several occasions throughout the manuscript authors refer that work is done but they do not explain how and what results are. Please include context or remove references.
7. Pg. 5 Tables 1&2 are taken from another work, please reference the works from where these results originate from.
8. Ln 219 – 221 – work aim and objectives this should be presented earlier, e.g. in the introduction and not at section 3.1 – please move it.
9. As mentioned earlies structure of the paper is confusing, please use journal template to structure the manuscript paper correctly.
10. Conclusion – 1st sentence – should be brought forward, this is first time authors really explain why they are doing the work. This should be brought forward in the paper.
11. Pg 16, ln 426: Too high voids – awkward writing, rewrite it, I am suggesting: High void content ….
12. Pg 16, ln 429: Define Dcd
Author Response
In accordance with all reviews, the article has been modified and edited. The abstract, research part and conclusions were edited in such a way that the objective, method of the experiment, results, conclusions and recommendations were clearly defined. The order of references has also been adjusted.
Authors' answers and comments:
- The English throughout the paper requires moderate corrections. Eg. Abstract, pg 1, ln 20: we made many tests.. – awkward writing I am suggesting: # of tests were carried out …. Answer: It has been edited where appropriate.
- Introduction paragraphs 1 and 2, here authors introduce risks and global climate challenges without a context, authors should present the challenge that they are addressing at start of the Introduction then this would make more sense as it is it reads discussion on the climate change rather than issue that authors are addressing. Answer: Climate change is also related to negative temperatures and their extremes. The text and context have been modified/added.
- Pg. 3, ln 91 – Authors state: “Asphalt binder is known to play an important role when determining fatigue performance of asphalt mixture.” Asphalt can’t determine asphalt fatigue properties. A ITFT or 4PBT us used to determine the fatigue properties, asphalt is material used to improve fatigue properties of the pavement… Answer: Corrected sentence wording.
- Pg 3 ln 100 – 103: Authors Refer to research programme carried out in Slovakia, please reference publicly available works that describe this work. Answer: From the 1970s to the 1990s, several reports were prepared at the Transport Research Institute on the effect of chemical spreading materials on pavements. We do not want to include any additional information in the article. Reference [31] was added.
- Pg 3, ln 112: computer decision tree [12] – please include schematic diagram of this tree and explain how it works. Answer: The authors do not consider it necessary to present a decision tree for the purposes of the article.
- On several occasions throughout the manuscript authors refer that work is done but they do not explain how and what results are. Please include context or remove references. Answer: The authors consider it appropriate to keep the references, the context is given. See references for specific results.
- Pg. 5 Tables 1&2 are taken from another work, please reference the works from where these results originate from. Answer: The tables are taken from the technical regulations valid in Slovakia and Decree No. 104/1997 Coll. valid in the Czech Republic. During the processing of the research project, two technical regulations TP 039 and TP 040 were in force in Slovakia. They stated the specifics listed in tables 1 and 2. From July 2022, these two regulations are unified into a common TP 039, where the temperature will change to -7°C when using NaCl (Tab. 1) and the total consumption of sprinkler salts must not exceed 80 g.m-2 in one intervention day. The data in the article has been updated.
- Ln 219 – 221 – work aim and objectives this should be presented earlier, e.g. in the introduction and not at section 3.1 – please move it. Answer: Modified/added.
- As mentioned earlies structure of the paper is confusing, please use journal template to structure the manuscript paper correctly. Answer: Modified/added. Journal template is used.
- Conclusion – 1st sentence – should be brought forward, this is first time authors really explain why they are doing the work. This should be brought forward in the paper. Answer: The research part and objectives are modified/added.
- Pg 16, ln 426: Too high voids – awkward writing, rewrite it, I am suggesting: High void content …. Answer: Corrected.
- Pg 16, ln 429: Define Dcd Answer: Added.
Round 2
Reviewer 2 Report
As far as I understand, it is just a comparison to see if low temperature affects the properties of the binder. Although the mentioned tests will be performed at room temperature, so they cannot provide a true picture of how freeze and thaw would affect the binders. Kindly improve your literature review by citing the latest references. e.g A review of the evolution of technologies to use sulphur as a pavement construction material Effective use of recycled waste PET in cementitious grouts for developing sustainable semi-flexible pavement surfacing using artificial neural network (ANN) Also redraw the conclusionsAuthor Response
In accordance with all reviews, the article has been modified and edited.
Authors' answers and comments:
As far as I understand, it is just a comparison to see if low temperature affects the properties of the binder. Although the mentioned tests will be performed at room temperature, so they cannot provide a true picture of how freeze and thaw would affect the binders. Kindly improve your literature review by citing the latest references. e.g A review of the evolution of technologies to use sulphur as a pavement construction material Effective use of recycled waste PET in cementitious grouts for developing sustainable semi-flexible pavement surfacing using artificial neural network (ANN) Also redraw the conclusions .
Answer: supplemented by related references, conclusion adjusted
Round 3
Reviewer 2 Report
NA